# Deep neural networks explain spiking activity in auditory cortex

Bilal Ahmed[1], Joshua D. Downer[2¤], Brian J. Malone[2¤], Joseph G. Makin [1]*

**1** Elmore School of Electrical and Computer Engineering, Purdue University, West Lafayette, Indiana, United States of America, **2** Otolaryngology and Head and Neck Surgery, UCSF, San Francisco, California, United States of America

¤ Current Address: Center for Neuroscience, U.C. Davis, Davis, California, U.S.A.
* jgmakin@purdue.edu

**Data availability statement:** All code are available via Github https://github.com/bilalhsp/deep-auditory-spikes/tree/main, and all data available via a public repository https://zenodo.org/records/16175377.

## Abstract

For static stimuli or at gross (∼1-s) time scales, artificial neural networks (ANNs) that have been trained on challenging engineering tasks, like image classification and automatic speech recognition, are now the best predictors of neural responses in primate visual and auditory cortex. It is, however, unknown whether this success can be extended to spiking activity at fine time scales, which are particularly relevant to audition. Here we address this question with ANNs trained on speech audio, and acute multi-electrode recordings from the auditory cortex of squirrel monkeys. We show that layers of trained ANNs can predict the spike counts of multi-units responding to speech audio and to monkey vocalizations at bin widths of 50 ms and below. For some multi-units, the ANNs explain close to all of the explainable variance—much more than traditional spectrotemporal receptive fields, and more than untrained networks. Non-primary neurons tend to be more predictable by deeper layers of the ANNs, but there is much variation by neuron, which would be invisible to coarser recording modalities.

## Author summary

Engineers have developed a variety of approaches to transcribing speech audio to text, or of automatically labeling the contents of images, but only those inspired by the architecture of the brain—artificial neural networks (ANNs)—are able to rival the performance of humans. ANNs are unbiological in many ways, but like the brain, they consist of very simple processing units organized into a hierarchy. Recently it has been shown that "units" in ANNs trained to solve such difficult tasks also become sensitive to the same features as neurons in the corresponding area of the cortex, with deeper layers of the ANN more closely resembling higher levels of the visual or auditory cortical hierarchy. Heretofore, however, the correspondence has only been shown for static stimuli, or at gross scales, either temporal (∼1s) or spatial (1000s of neurons). Auditory stimuli, in particular, elicit neural responses that encode information at the millisecond level.

**Funding:** This work was supported by the NIH 1R01DC021600-01 (to JGM) and a Ralph W. and Grace M. Showalter Research Trust Award (to JGM). The funders had no role in study design, data collection and analysis, decision to publish, or preparation of the manuscript.

**Competing interests:** The authors have declared that no competing interests exist.

Here we show that ANNs trained on speech audio develop representations that can predict spiking activity in the auditory cortices of squirrel monkeys with high accuracy at time scales of tens of milliseconds.

## Introduction

The primate auditory system transforms incoming acoustic information dispersed in time and frequency into distinct auditory "objects" that can be interpreted, localized, and integrated with information from the other senses. A human listener can resolve, for example, a monophonic musical recording into piano and guitar, or again into more abstract "objects" like a melody over a sequence of chords. Although two decades of electrophysiology have yielded precise characterizations of many of the tuning properties of neurons in auditory cortex, we currently do not understand how these together underwrite such computations.

On the other hand, we now have in deep artificial neural networks (ANNs) a more easily investigated system that—as of this last decade—solves such problems at human performance levels. Although crude as biophysical models, ANNs strongly resemble biological neural networks in terms of *computation* and *representation* [1–5]. Recently, this has been demonstrated convincingly for the ventral visual pathway [1,3,4]: the deep layers of ANNs trained as image classifiers explain more than half the variance of single-unit responses in areas V4 and IT atop the visual hierarchy, more than any other modeling approaches [6]. This is despite the fact that these ANNs were not trained to predict activity in the brain, but only to complete an ecologically relevant "pretext" task (image classification); the connection between the artificial and real neural activity was made only by way of a few hundred parameters of a linear readout map.

In addition to visual cortex, ANNs have been studied as encoding models for auditory cortex, as in the present study, chiefly for fMRI data [2,7–12]. The major findings from these studies are threefold: (1) ANNs trained on challenging audio pretext tasks (like automatic speech recognition) predict the BOLD signal in auditory cortex better than the spectrotemporal filters in classical models; (2) there is some correspondence between the hierarchical organizations of the ANNs and of auditory cortex; and (3) model prediction quality correlates with performance on the pretext task, but more strongly for some pretext tasks than others. For example, models trained to recognize speech in noise are better predictors of the BOLD signal than those trained to recognize clean speech; and models trained on multiple pretext tasks are better predictors than those trained on just one [12].

The advantage of fMRI over invasive techniques is that it allows for human subjects. On the other hand, fMRI is necessarily limited to predicting the average cortical activation (voxel intensity) over the course of the entire two-second stimulus, since it cannot resolve temporal fluctuations faster than about 1 Hz. Therefore although the studies just cited employ ANNs operating on fine time scales, their outputs are simply averaged across approximately 1 second before making predictions. This limitation is likely to be particularly destructive for the auditory system and its stimuli, which are information-rich in precisely the temporal (as opposed to spatial) dimension. Population-based decoding in core auditory cortex, for example, is optimal at a temporal resolution of less than 2 ms for discriminating sounds based on their temporal features [13].

Very recent work from the Chang lab has examined ANNs as models for electrocorticography (ECoG) in the auditory cortex of humans listening to speech [14], and found something like the three points adduced above for a few different ANN architectures. For example, training ANNs on a Mandarin, rather than English, pretext task yields better predictions of

human neural responses to Mandarin—in native speakers of Mandarin, but not of English. Model predictions of the ECoG envelope in the high-$\gamma$ range were made at a fine time scale, effectively up to about 20 Hz (beyond which the analytic amplitude has no power).

What remains unclear is whether the correspondence between ANNs and auditory cortex continues to hold at the level of spiking activity. Individual ECoG channels, like fMRI voxels, represent the aggregate activity of $\sim 10^5$ neurons, so as far as these results go, the correspondence might emerge only at the population level. To explain the tuning properties of auditory cortical neurons, it is necessary to record spiking activity.

Accordingly, we made multielectrode recordings from single- and multi-units in the core, belt, and parabelt areas of the auditory cortex of squirrel monkeys, during which the animals were exposed to a battery of $\sim 600$ spoken English sentences and $\sim 450$ monkey vocalizations. We then compared these to the responses (to the same battery of stimuli) of units in ANNs that have been trained on speech data—either fully supervised (mapping audio to letters or characters) [15–17], "weakly" supervised [18], or with a combination of supervised and unsupervised learning [19]—and with a range of different architectures and model sizes: fully convolutional, recurrent, and self-attentional. In particular, we asked how well sequences of spike counts at fine time scales ($\sim 50$-ms bins) can be (linearly) predicted by different layers of the ANNs.

For the best network, the median correlation between model predictions and cortical responses for held-out data exceeds $\sim 0.7$ (after correcting for unpredictable trial-to-trial variation and removing unpredictable neurons); typically increases with layer until approximately midway through the network, although this varies by recording channel and by network architecture; and exceeds that of untrained ANNs and of classical models based on spectrotemporal receptive fields (STRFs) [20]. We also find some correspondence between the hierarchies of our ANNs and the traditional primary (core) vs. non-primary (belt, parabelt) distinction of primate auditory cortex. Above all, these results provide strong evidence that the representational correspondence between (on the one hand) ANNs and (on the other) cortical responses to time-varying stimuli holds down to the level of spiking activity at fine time scales, which has not been established heretofore.

## Materials and methods

### Ethics statement

All animal procedures were approved by the Institutional Animal Care and Use Committee of the University of California, San Francisco (protocol numbers AN109411-03 and AN173817-01), and followed the guidelines of the National Institutes of Health.

### Data collection

Electrophysiological signals were recorded from the auditory cortex of squirrel monkeys using a preparation described in detail elsewhere [13], which we summarize briefly here. Over the course of several months, recordings were made from penetrations into the core, belt, and parabelt areas of right auditory cortex of three animals (B, C, F). For monkey C, recordings were also made in left auditory cortex. We used a variety of probes with 1, 16, 32, 48, or 64 channels, selected for each recording session based on which probe was most suitable for accessing the target area on that day. As a result, probe layouts varied across sessions (see S2 Table for details). Our ideal approach was to advance the probe until the deepest channel no longer registered neural activity. While this generally allowed us to sample across all cortical layers at most recording sites, accessing cortical fields near the edges of the craniotomy often

limited our ability to record from all channels on a given probe. Additionally, the electrode insertion angle was not always orthogonal to the auditory cortex, making it difficult to confidently assign specific channels to specific layers. Based on these constraints, we infer that the majority of the recorded neurons presented here are located within the supragranular and granular layers of the auditory cortex.

Auditory stimuli were played from a free-field speaker inside the soundproof chamber where recordings took place. During each recording session, two sets of stimuli were presented. The first set consisted of sentences from the TIMIT corpus, each approximately 1–3 seconds in length, which we have found to elicit strong responses from the auditory cortex of squirrel monkeys. A total of 499 unique sentences were presented: 489 were presented exactly once (in random order), while the remaining 10 were repeated 11 times each, for a total of 110 presentations (also in random order). The second set consisted of monkey vocalizations, each approximately 1 second in length, including grunts, screams, and coos. In this set, a total of 303 unique vocalizations were presented: 292 were presented once, and the remaining 11 were presented 15 times each, for a total of 165 presentations, also in random order.

Spikes were identified based on threshold crossings, defined as voltage deflections exceeding 3.5 standard deviations from the mean extracellular voltage on each probe. Unthresholded data were collected using Intan software and subsequently thresholded using custom MATLAB code. For the main analyses, spikes were counted in 50 ms bins; for Fig 4, spikes were binned at 20 ms. Because spike-sorting was not feasible, we work with multi-unit activity (MUA) throughout this study, and we refer to the source of spikes on a single channel as a "multi-unit."

## Trial-to-trial neural variability

Neurons do not respond identically to identical stimuli, whether because they are tuned instead or additionally to other, uncontrolled stimuli; as a consequence of feedback or attention; due to false alarms or misses in spike detection; or simply because of intrinsic noise in the spiking process.

To characterize this variability, we analyzed the response of each multi-unit to the repeated stimuli (see previous). In particular, since sample correlation coefficients converge on the true correlation as the number of samples (in our case, bins) increases, we first concatenated together responses from multiple trials. More precisely (see S1 Fig), for each of the $M$ unique test stimuli, we randomly selected a single pair of responses, $(U_m, V_m)$, with $U_m \neq V_m$, for $m = 1, \ldots, M$. We then concatenated together all $M$ of the $U_m$ into a single "long" sequence, $U$, and likewise for all $M$ of the $V_m$ into a sequence $V$, in the same order. Thus for TIMIT, for which the repeat set was 10 unique sentences, each on average 1.6 seconds long, the two concatenated sequences $U$ and $V$ were each about 320 samples long (20 samples/second × 1.6 seconds/sentence × 10 sentences). For the monkey vocalizations, $U$ and $V$ had about 220 samples each (20 samples/second × 1.0 second/vocalization × 11 vocalizations). The sample correlation between $U$ and $V$ provides an estimate of this multi-unit's reliability. To generate more such estimates, we repeated this entire procedure 100,000 times, yielding 100,000 correlation coefficients for each of the ~1700 multi-units (recording channels).

To quantify this variability, we also constructed a null distribution of trial-to-trial correlations for each recording channel, analogous to the true distribution described above. The key difference is that, for the null distribution, we circularly shift the $V$ responses before computing the correlations. This procedure preserves the marginal response statistics for each channel while disrupting any temporal alignment or correlation with $U$. We constructed a separate true and null distribution for each recording channel for each of the stimulus sets

(speech and monkey vocalizations). These distributions were computed for 50 ms and 20 ms for, respectively, the main results and the analysis in Fig 4. We then put these distributions to three related but distinct uses.

**Identifying tuned units**  For each multi-unit, we asked whether its distribution of trial-to-trial response correlations is significantly greater than the null distribution under a Wilcoxon rank-sum test. With a $p$-value of 0.05, this yields 1195 and 1231 tuned neurons, at 50-ms bin widths, for speech and vocalizations, respectively (see S3 Table).

**Restricting to well-tuned units**  However, units that are classified as "tuned" under this criterion can still have highly overlapping distributions of true and null trial-to-trial correlations (see S1 Fig). We therefore adopted a stricter notion of tuning in our analysis, requiring the means of true and null distributions to be separated by a gap of at least $\delta = 0.5$ standard deviations of the null distribution. We chose this point as it corresponds roughly to the elbow in the plot of number of accepted units vs. the gap between null and true distributions; see S1B Fig. (However, the results are robust to this choice. In the Supporting information, we reprise the major results using $\delta = 1.0$; see S8 Fig–S10 Fig.) At 50-ms bins, there are 404 and 489 such multi-units for English speech and monkey vocalizations, respectively, out of a total of 1718. We call such units "well-tuned."

**Correcting for unexplainable variance**  Since trial-to-trial neural variability to the same stimulus cannot be explained by any encoding model, it is common to "correct" model-neuron correlations for this excess variance. For each neuron or multi-unit activity, one first estimates the correlation between responses to identical stimuli, and then normalizes the model-neuron correlation by (a function of) this number. More precisely, there are two sources of noise in the trial-to-trial correlations (since there is independent noise on the two trials) but only one in the model-neuron correlations (since the model is noise-free); so the model-neuron correlation must be normalized by the *square root* of the correlation between responses to identical stimuli; see S1 Text: Noise correction of correlations for a derivation. We estimated this correlation with the mean of the trial-to-trial response correlations just described. This is very similar to (e.g.) the noise correction used by Pennington and David [21].

## Encoding models

**ANN architectures and pretext tasks**  We considered six different neural-network architectures, all designed for speech-to-text tasks:

- WAV2LETTER [15]: 15 convolutional layers, mapping raw waveforms to letters. We modified the original network to have more slowly growing receptive fields (by reducing the convolutional kernel widths and strides), which is more computationally expensive but arguably more biologically plausible. In order to reach the same final sampling rate, we used 15 rather than 12 layers. We trained the model on 960 hours of LibriSpeech, a large corpus of spoken (read) English speech, until character error rates fell below 8% on a held-out test set.
- SPEECH2TEXT [16], based on the Huggingface implementation [22]: two convolutional layers followed by 12 transformer-encoder (self-attentional) layers, mapping log-mel filter bank to word pieces. We do not analyze the decoder. We used a model (pre)trained with standard supervised learning on LibriSpeech.
- WAV2VEC2 [19], based on the Huggingface implementation [23]: seven convolutional layers (with layer normalization [24] and GELU activation functions [25]) followed by one convolutional layer of positional embedding, and then 12 transformer-encoder

layers, mapping raw waveform to letters. Our model was (pre)trained under a self-supervised contrastive loss and then only the transformer layers (and a linear projection layer following transformer layers) were fine-tuned with supervision on a speech-to-text task. The contrastive loss obliges the transformer layers to generate output that can be used to distinguish tokens masked out of its own input sequence (the output of the convolutional network) from tokens taken from other input sequences.

- DeepSpeech2 [17], based on a publicly available implementation [26]: two convolutional layers followed by five bidirectional recurrent (LSTM [27]) layers, mapping amplitude spectrogram to letters. We used a model (pre)trained with standard supervised learning on about 12,000 hours of read and conversational English speech (including LibriSpeech, Switchboard, WSJ, and Fisher).

- Whisper (tiny) [18] (based on the trained instance uploaded to Huggingface by OpenAI [28]): two convolutional layers followed by four transformer-encoder layers, mapping log-mel filter banks to word pieces. We do not analyze the decoder. The model was trained to recognize speech from audio under "weak" supervision, that is, with possibly low quality labels sourced from the internet, but (consequently) upwards of 700,000 hours of audio data. About 20% of the data were non-English speech, and training also included a translation task, in addition to standard ASR.

- Whisper (base) [18] (based on the trained instance uploaded to Huggingface by OpenAI [28]): two convolutional layers followed by six transformer-encoder layers, mapping log-mel filter bank to word pieces. We do not analyze the decoder. The model was trained the same way as Whisper (tiny).

These choices allowed us to investigate the effects of

- network input: waveform (wav2letter, wav2vec2) vs. spectrogram (the remainder);
- network architecture: pure convolution (wav2letter), self-attention (wav2vec2, speech2text, Whisper), and recurrent (DeepSpeech2);
- training: "weak" supervision (Whisper), a combination of supervised and unsupervised learning (wav2vec2), or fully supervised (the remainder);
- network size (Whisper tiny vs. base)

See also S2 Text: Pretrained networks and S4 Table for more details.

**Untrained ANNs**   In order to distinguish the effect of training (on the pretext task) from the effects of architectural choices, we also considered the untrained counterparts to these networks. Untrained networks were obtained by (re-)initializing the network weights using the default initialization schemes for each network.

**Predicting spiking activity with ANNs**   The effective sampling rate in the ANN (imposed by the "strides" of the convolutions) declines across layers. Therefore, before fitting linear readouts maps, all layer activities were resampled to 20 Hz to match the 50-ms bins for the main results of this study (and for the analyses reported in Fig 4, to 50 Hz). We then fit a temporal receptive field (TRF) from each ANN layer to each well-tuned multi-unit (see Fig 1). In particular, the predicted neural response at one bin was allowed to depend linearly on a 250-ms window of ANN activity. TRFs were fit on the non-repeated stimuli (see Data collection above) by minimizing $L_2$-regularized squared error (~20,000 samples and ~6000 samples at 50-ms bins for speech and monkey vocalizations, respectively). The magnitude of the $L_2$ penalty was determined with three-fold cross validation over a range of regularization parameters ($10^{-5}$ to $10^{15}$). All fits were made with the `naplib` python package [29], partly customized to run on GPUs.

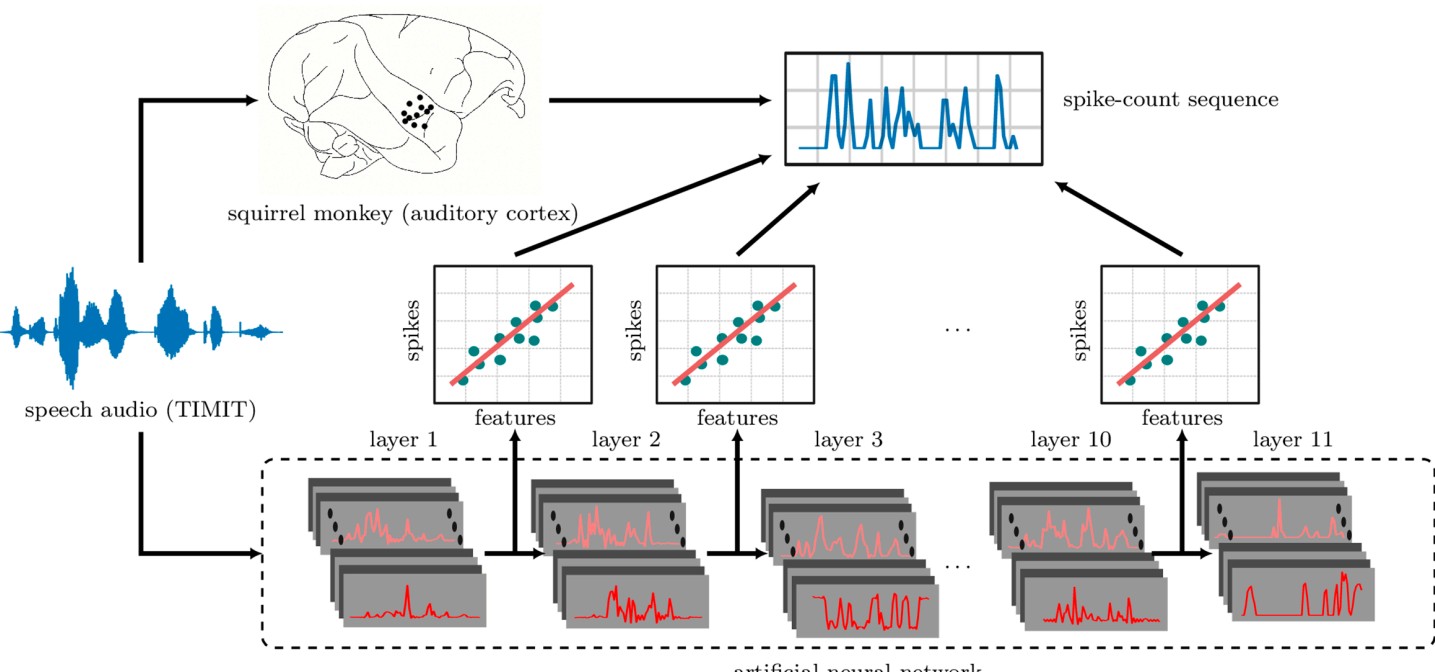

**Fig 1. Schematic of ANN-based encoding models.** Spiking activity (upper right) in response to stimuli (e.g., speech; waveform at left) is recorded from squirrel monkey auditory cortex (black dots on brain in upper left). The exact same stimuli are presented to the ANN (bottom). The spike sequences are then regressed separately (schematic linear plots, middle) onto each hidden layer's responses to the same stimuli, yielding a temporal receptive field (TRF) for each layer of the ANN. Performance is evaluated on held-out stimulus-response pairs by computing the correlation between the TRF-based predictions and the spike-count sequences.

We emphasize the rationale for using a (merely) linear readout: We want to be able to identify the layer of the network that best explains cortical neural activity. Allowing (arbitrary) nonlinear maps would break this correspondence, since the layers themselves are related by nonlinear maps. (We can say, colorfully, that if layer $n$ makes the best predictions through a linear map, then layer 1 also makes the best predictions through the nonlinear map consisting of the neural network itself up to layer $n$, and then out through the linear map.) At the other extreme, we are uninterested in mapping single ANN units to single electrode channels because a layer is arbitrary up to a (full-rank) matrix multiplication (the subsequent weight matrix could absorb the inverse of this matrix).

**Predicting spiking activity with STRFs.** As a baseline, we also fit spectrotemporal receptive fields (STRFs) [20] to every well-tuned multi-unit. The fitting process was identical to the one just described for the ANNs, except that the input to the TRF in this case is not the ANN layer activity but the time-varying amplitude spectrum of the stimulus.

We compared six different methods for computing the amplitude spectrum: a wavelet transform designed to mimic cochlear processing of natural sounds [29], a cochleagram representation computed using filters spaced according to the equivalent rectangular bandwidth (ERB) scale, and four spectrogram-based methods, including the mel-spectrogram computed by the Python package Librosa, and the spectrograms employed by the ANN models we studied—specifically, those used in DeepSpeech2, speech2text, and Whisper. Among these, the cochleagram method yielded the highest model-neuron correlation performance (see S2 Fig). Based on this observation, we adopted the cochleagram as our default spectrogram representation for computing STRF-neuron correlations in the main analyses. The

cochleagram was computed following [12]; specifically 211 filters were applied in the range of 50-8000 Hz, with an overlap of 87.5%. The resulting cochleagram was downsampled to 20Hz before fitting the linear readout to predict neural data. See S5 Table for a detailed comparison of the spectrogram computation methods.

### Evaluating model predictions

Our main metric of model performance is the noise-corrected correlation coefficient (see above), which we computed for every combination of well-tuned units and encoding model. Note that an "encoding model" corresponds to a single layer from a single ANN, either trained or untrained, or to a STRF. Correlation coefficients were computed between the sequences of predicted and actual (biological) responses to the 110 and 165 held-out stimuli for speech and monkey vocalizations, respectively. Recall that these correspond, respectively, to 11 repeats of 10 unique sentences and 15 repeats of 11 unique vocalizations.

Since longer sequences yield better estimates of the correlation coefficient, we first assembled responses into "long sequences" (similar to the construction of the trial-to-trial correlations; see above). That is, for each multi-unit, we concatenated together the responses to the first instances of each of the 10 sentences; and likewise for the second instances; and so on up to the eleventh instance, yielding 11 long sequences of responses. For each model (ANN layer or STRF), we assembled a single long sequence of responses to these 10 unique sentences. We then computed the correlation between that model's long sequence and each of a particular multi-unit's 11 long sequences. For each multi-unit, we report the mean of these 11 correlation coefficients. (We used the same procedure for the responses to monkey vocalizations, *mutatis mutandis*.)

To compare models with each other, we compare the distributions of correlation coefficients across all well-tuned units. Since the distributions to be compared consisted of paired samples (each corresponding to the same multi-unit), but are not Gaussian, we tested for significant differences with the Wilcoxon signed-rank test. For each layer of an ANN, we make two comparisons: one with its untrained counterpart and another with the STRF, resulting in a total of $2 \times L$ comparisons for an $L$-layered model. To control the false discovery rate, we apply the Benjamini–Hochberg correction separately for each model.

### Evaluating performance on the ecologically relevant pretext task

To relate the ANNs' performances at predicting neural activity to their performances on the pretext task, we evaluated the word-error rates (WER) of all the six ANNs on the following datasets (see S6 Table):

- TED-LIUM 3 [30]: Test split of release 3 from the TORCHAUDIO repository [31].
- Common Voice 5.1 [32]: Test split of Common Voice 5.1 for English language [33].
- VoxPopuli [34]: Test split of transcribed speech for English language from the Meta Research repository [35].

## Results

The basic element of our analyses is the thresholded voltage on a single channel. These are likely to include multiple neurons; we refer to these as "multi-units." (We discuss spike sorting in Materials and methods.)

Fig 2A (bottom left) shows the response of a multi-unit in monkey B to ten repetitions of a single sentence, whose spectrogram is shown on the top left. (No instance of this sentence

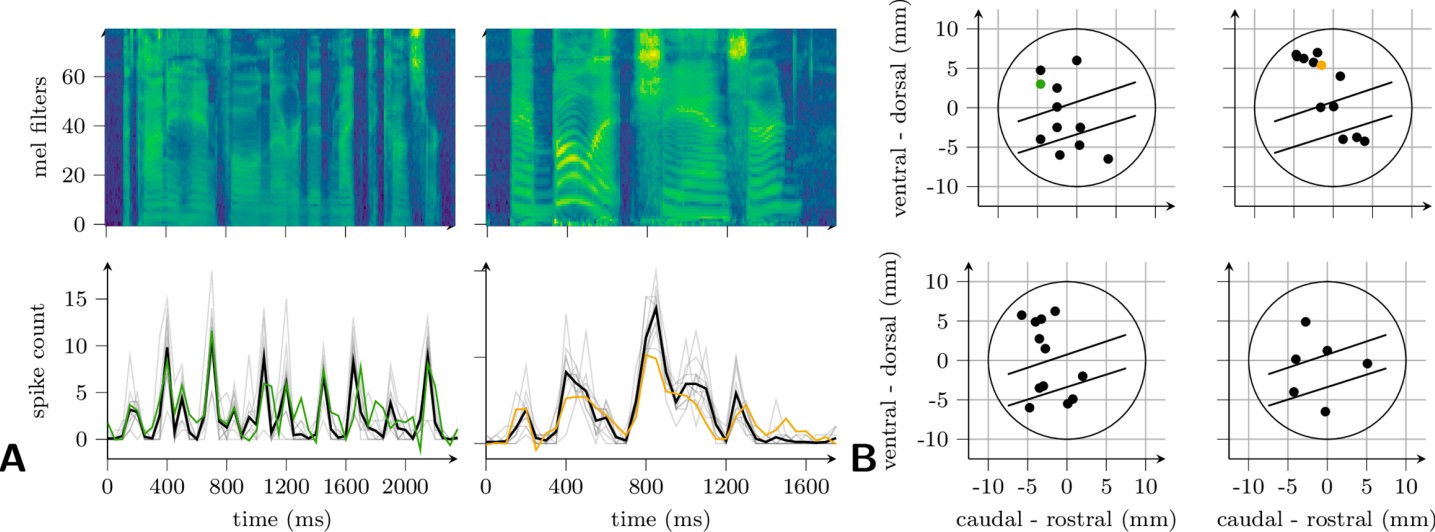

**Fig 2. Example cortical responses and ANN predictions.** A: (Left) Sequences of spike counts (50-ms bins) in response to an English sentence ("A tiny handful never did make the concert"; spectrogram above). Below are cortical responses to ten repetitions (gray) and their mean (black), and the prediction from the Whisper [base] ANN (green). (Right) the same as the left panel but for a different sentence ("A bullet, she answered"), ANN (wav2vec2, in yellow), and monkey. The locations of the recordings are indicated by color in B. B: Locations of recording sites across monkeys C (top row), B (bottom left), and F (bottom right). All recordings in right hemisphere except top left panel. The large circle indicates the location of the recording cylinder, within which the upper half plane corresponds roughly to primary auditory cortex (core); the lower half, non-primary (belt and parabelt). The approximate location of belt in each cylinder is between the two parallel lines.

was used in fitting the linear map from ANN to neural response; see Materials and methods). The sequence of spike counts (in 50-ms bins) predicted by layer 2 of the Whisper [base] model is superimposed in green on top of the sequences of actual spike counts from these ten trials (gray) and their mean (black). The location of the electrode in core auditory cortex that recorded this activity is shown in green in Fig 2B. The right half of Fig 2A shows similar results for a recording site in a different animal (colored yellow in Fig 2B), in response to a different sentence, and predicted by a different network (wav2vec2, layer 8). In both cases, the match is evidently quite close even at fine time scales.

## Performance of ANNs as models of auditory cortex

We expand our view to all six ANN architectures, all stimuli, and the entire set of tuned multi-units. We identified a multi-unit as tuned if its responses to multiple tokens of the same stimulus correlate with each other above chance (see Materials and methods). Note that this determination is sensitive to the width of the bin in which spikes are counted; unless stated otherwise, we used 50-ms bins. About 70% (1195/1718) of multi-units were tuned to speech under this criterion, and 72% tuned to monkey vocalizations (see S3 Table); this validates the use of speech audio as a stimulus. Nevertheless, many of these nominally tuned multi-units have trial-to-trial correlations quite close to chance (see S1 Fig), so we restrict our analyses to a subset of well-tuned multi-units (see Materials and methods), yielding 404 for speech and 489 for monkey vocalizations.

To make predictions, we fit a separate linear temporal receptive field (TRF) from each layer of each ANN to each multi-unit activity (across all electrode channels, all recording sites, and all three monkeys), i.e., to each sequence of spike counts in 50-ms bins. As a baseline, we also fit a spectrotemporal receptive-field (STRF) to each multi-unit activity. To evaluate these

encoding models, we compute the correlation between sequences of spike counts and corresponding TRF predictions on a held-out set of stimuli. When reporting these correlations, we follow the standard practice of correcting for unexplainable trial-to-trial variability [36]. (An alternative [14] is to report the ratio of each ANN-neuron correlation to the corresponding STRF-neuron correlation. This has the advantage of canceling out the noisy normalizer in the correction, although it obscures the absolute correlations achieved. We re-present Fig 3 this way in S3 Fig.)

Fig 3A (colored shading and colored lines) shows the distributions of correlations across all multi-units and at each layer of each ANN, in response to English sentences. First, we note that, at their intermediate layers, all ANNs make superior predictions to the STRF (median correlation shown as gray bar in leftmost plot and as a dashed line in all other plots); i.e., the distributions of model-neuron correlations are significantly higher (gray stars; Wilcoxon signed-rank test, $p < 0.01$). Indeed, the most predictive ANN layers correlate with the median multi-unit at about 0.6 (noise-corrected). This is comparable to the best models of auditory neurons to date [21], which are complex neural networks fit directly to the neural data, whereas our models fit only the linear readout. Nevertheless, we caution that direct comparisons are difficult, since that study and ours differ in the animal model (ferret in theirs), number of data collected, neuron-inclusion criteria, and other details of the experiments.

Second, the distribution of correlations typically rises over the first few layers, peaks before the middle of each network, and then falls off. The rise is expected: For networks that take the raw waveform as input (WAV2LETTER [15] and WAV2VEC2 [19]), it is not surprising that multiple layers of nonlinearities provide for better predictions, since the relationship between

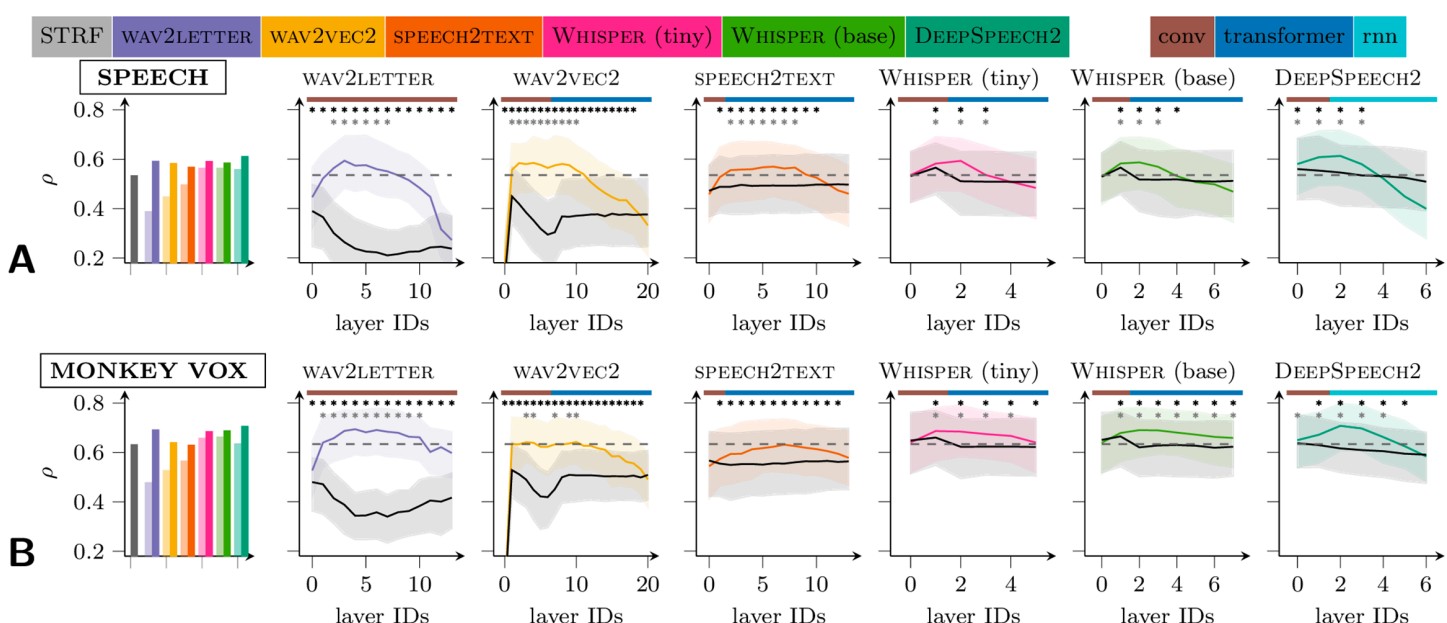

**Fig 3. Model-neuron correlations.** All subpanels show correlations between model predictions and the multi-unit activity they are supposed to predict. A: Model-neuron correlations for speech (TIMIT) stimuli. Bar plot: median (across multi-units) correlation of the STRF (gray) and of the best layers of each of the neural networks, both trained (dark colors) and untrained (light colors). Line plots: the distributions of correlations as a function of ANN layer for each of six trained networks (colored) and their untrained counterparts (gray). The median (solid line) and interquartile range (shaded region) are shown. At starred layers, the trained ANN is significantly superior to its untrained counterpart (top row of stars) or a STRF (bottom row of stars; Wilcoxon signed-rank test with $p < 0.01$). To control the false discovery rate, we apply the Benjamini–Hochberg correction separately for each model. The layer type is indicated along the top of each plot: convolutional (brown), self-attention (blue), and recurrent (light blue). B: The same as A but using monkey vocalizations for stimuli.

auditory-cortical responses and sound is known to be highly complex and nonlinear. But the same holds even for networks that take the spectrogram as input (SPEECH2TEXT [16], the WHISPER models [18], and DEEPSPEECH2 [17]), which implies that even the STRF (which is linear in the spectrogram) is several nonlinearities away from the optimal response model. The post-peak fall off in correlations could be a result of ANN specialization for the speech-recognition task: the deepest layers predict phonemes or characters, a task that is arguably foreign to the squirrel monkey. Indeed, if we (pre-)train WAV2VEC2 on AudioSet [37], a large collection of sounds from internet videos, and then skip supervised fine-tuning altogether, the post peak fall-off mostly disappears (S4 Fig). We return to this theme in the Discussion.

## The importance of task optimization

To verify that task optimization improves model predictions, we investigate the predictive performance of *untrained* networks. Because each layer of an artificial neural network contains nonlinearities, a linear readout from deeper layers can be more expressive, even when the network is untrained. To determine how much of the ANN's predictive power is due to training on the task of automatic speech recognition (ASR), and how much merely to these stacked nonlinearities, we also attempt to predict cortical activity with untrained networks. In particular, we "reset" the weights to their initial values, i.e. before training began, and then fit a new set of linear maps (TRFs) from each layer to each cortical unit (see Materials and methods). The distributions of the resulting correlations are also shown in Fig 3A, in grayscale. Trained networks are superior to untrained networks almost everywhere (Wilcoxon signed-rank test; $p < 0.01$ indicated with top row of black stars), and the most predictive layers are all in trained networks.

In networks that take the spectrogram as input, the difference between trained and untrained networks is much smaller than in networks that take the waveform as input—presumably because in the latter, something like the spectrogram computation is learned by the trained network, but cannot be randomly assembled by the untrained network. This accords with the classical emphasis on STRFs and with the well understood transformations of the audio signal by the cochlea.

## The ecological relevance of speech recognition

In order to avail ourselves of ANNs trained with supervision on large datasets, we have let the pretext task for our networks be speech recognition. This in turn has narrowed our focus (up to this point) to English-sentences stimuli, since it is unclear how such networks will respond to sounds not occurring in their training sets. Three findings support this choice: (1) Roughly as many multi-units—indeed, mostly the same multi-units—are well tuned to English and to monkey vocalizations; (2) our ANNs linearly predict the spiking responses to English sentences about as well as the best nonlinear models of auditory cortex [21]; and (3) the pretext training (on speech) improves model-neuron correlations beyond what can be achieved with random nonlinearities. Although English speech is not entirely irrelevant to these animals, these results together suggest that neurons in squirrel-monkey auditory cortex are tuned to audio features that are higher-order than spectrograms but generic enough to be useful (if not used) for speech recognition, as well as processing other sounds.

If this is the case, we also expect our ANNs to explain the responses of auditory neurons to monkey vocalizations. Fig 3B shows the results (in the same format as Fig 3A). Note that the linear readout was re-fit for these stimuli, but the ANNs are identical to the ones analyzed in Fig 3A: they have been pretrained only on speech tasks. We observe three important points:

- Like the responses to speech, responses to monkey vocalizations are best explained by the intermediate layers of the trained ANNs, which are likewise always better than their untrained counterparts. The best overall predictions are made by layer 2 of DEEP-SPEECH2, just as for speech stimuli.
- The responses to monkey vocalizations are more easily predicted than the responses to speech, despite the ANNs being trained on a speech task. However, this also holds for STRFs, suggesting that monkey vocalizations are in some sense simpler stimuli than speech.
- Consequently, the weakest networks, WAV2VEC2 and SPEECH2TEXT, provide little to no improvement over STRFs in predicting responses to monkey vocalizations. But the stronger networks still do.

Still, there is a limit to the ecological relevance of speech to the squirrel monkey. The ANNs' performance in speech recognition does not, for example, correlate with model-neuron correlations (S5 Fig), as has been observed in studies in humans [2,12,14]. And although in core we found about as many multi-units well tuned to speech as to monkey vocalizations, we found only about 60% as many in non-primary areas (see S3 Table) (see discussion).

## The time scale of predictions

Up to this point, we have performed all analyses (Fig 2 and 3 as well as S5 Fig and S3 Table) with 50-ms bins, i.e. at 20 Hz. This represents a modeling decision: the various layers of the various ANNs operate at various sampling rates, which we have simply resampled to a single rate (20 Hz) in order to match a single bin size for counting spikes (50 ms), which is itself arbitrary. At the extreme, the biological and artificial activities could be summed over the entire stimulus-presentation period, and just one prediction made per stimulus. This is the approach that has (necessarily) been taken in the fMRI studies described previously [2,7–12]. However, the large values of correlations in Fig 3 and the examples in Fig 2A strongly suggest that the ANNs in this study can predict neural activity at much finer time scales than this. Here we ask more precisely what resolution yields the best correlations.

To answer this question, we low-pass filter the ANN activities with various cut-off frequencies before attempting to predict spikes binned at "high frequency" (50 Hz). If increasing the cut-off frequency—e.g., above 10 Hz—increases model-neuron correlations, then the model is capable of explaining aspects of the neural activity that are faster than 10 Hz. Fig 4 shows the result for a range of cut-off frequencies up to the Nyquist limit, for all six (trained) ANNs. More precisely, at each cut-off frequency, we plot the distribution of model-neuron correlations for speech stimuli from the layer that is most predictive with 50-ms bins. The pattern is the same for all networks: model-neuron correlations increase monotonically with cut-off frequency. This shows that the ANNs are predicting fine temporal structure in the spike-count sequences, all the way up to their Nyquist limits.

## Hierarchy

We now investigate the relationship between the hierarchies of auditory cortex and of the deep neural networks used to explain it. So far we have not distinguished between neurons recorded from core areas and those recorded in non-primary (belt, parabelt) regions of auditory cortex. Here we ask whether a multi-unit's location in auditory cortex (and, by implication, the auditory hierarchy) is related to the depth of the ANN layer that best predicts its responses to speech and monkey vocalizations.

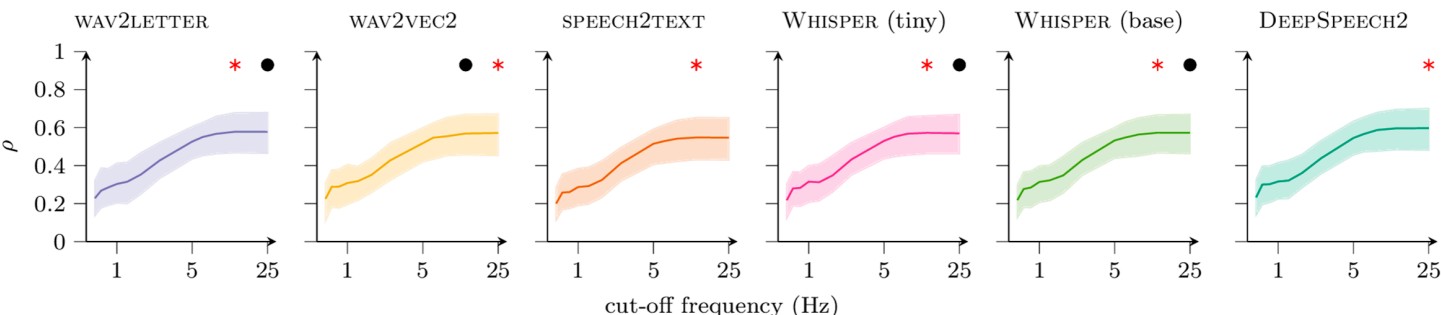

**Fig 4. Distributions of model-neuron correlations as a function of maximum frequency of the model predictions.** Median (solid line) and interquartile range (shaded) are indicated. For each ANN, results are shown only for the layer that was most predictive at 50-ms bins. First, the layer's responses were low-pass filtered at the frequency indicated on the horizontal axis. Then a linear readout (TRF) was fit to predict spiking activity binned at 20 ms. The vertical axis shows the resulting distribution of correlations on the test set. The red star indicates the cut-off frequency yielding the largest median (across multi-units) correlation; black dots indicate frequencies yielding correlation distributions indistinguishable from that at the red star (Wilcoxon signed-rank test with $p < 0.01$).

For each well-tuned multi-unit and each ANN, we find the depth of the most predictive layer, as a fraction of the total network depth. Fig 5 shows the distribution of these depths across all networks, broken out by primary (blue) and non-primary (orange) multi-units. (We break these out by network in S6 Fig.) Whether the stimuli are speech or monkey vocalizations, non-primary MUA is significantly more likely than primary MUA to be best predicted by deeper layers (Wilcoxon rank-sum test, $p < 0.001$). Nevertheless, the primary and non-primary distributions are broad compared to the differences between them. This suggests a "soft" anatomical hierarchy, with many neurons in belt and parabelt playing the role of lower-level neurons, and vice versa.

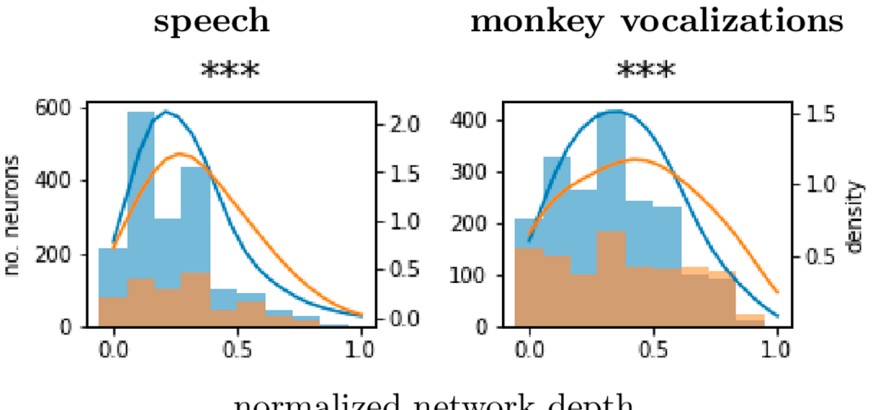

**Fig 5. Distributions of most predictive layers (normalized) for primary (blue) and non-primary (orange) multi-unit activity.** Histograms and corresponding kernel density estimates are shown as a function of network depth (from shallowest to deepest), pooled across all neurons and all six ANNs. The distribution of preferred layers is significantly "deeper" for non-primary than primary neurons (Wilcoxon rank-sum test, $p < 0.001$).

## Discussion

We have shown that deep neural networks make good encoding models for spiking activity in the auditory cortex of squirrel monkeys, even though those networks were trained, not to predict the activity of those neurons, but instead to solve a challenging auditory task, automatic speech recognition. This is (to our knowledge) the first time such an approach has been used successfully to explain spiking responses to dynamic stimuli at high temporal resolution—in this case, bin widths for counting spikes as small as 20 ms.

Since the spike-count predictions are allowed to depend on a temporal *window* of layer activity (see Materials and methods), each ANN-based encoding model is a temporal receptive field (TRF) in the feature space of a particular ANN layer, just as the STRF is a TRF in the amplitude spectrum. Since the most predictive ANN layers provide better predictions than STRF models for most multi-units, we can say that neurons in primate auditory cortex are more tuned to the intermediate features of deep neural networks than to the amplitude spectrum. Indeed, these features explain essentially all of the explainable variance of some of the multi-units from which we obtained recordings.

### Untrained ANNs and STRFs as encoding models

Nevertheless, although worse than the best trained ANNs, STRFs and untrained ANNs provide perhaps surprisingly good encoding models. We believe this has sometimes been obscured in the literature, due to the different ways of constructing STRFs and "untrained" models. In short, the optimal STRF should predict neural responses better than untrained ANNs that take the waveform or a suboptimal spectrogram as input; but the same as or even worse than untrained ANNs that take the optimal spectrogram as input (recall Fig 3). The rationale is as follows.

In our study, untrained models were produced by (re-)initializing the network weights and biases according to their default initialization schemes. These typically set the weights to small random values (and biases to zero), which produces full-rank matrices with high probability. Furthermore, the activation functions in these ANNs are often linear (or even identity functions) near zero. Since the random weights are small and symmetrically distributed about zero, their product with inputs is also typically near zero, and therefore passes through the linear portion of the nonlinearity. So the "nonlinearities" in untrained networks are mostly information-preserving linear transformations.

Consequently, the first few layers of untrained networks that take the spectrogram as input are likely to linearly predict neural activity as well as the spectrogram itself, i.e. as well as a STRF. This is just what we find for the WHISPER networks and DEEPSPEECH2 (Fig 3). It is *not* what we find for SPEECH2TEXT, whose early layers are much worse than the (optimal) STRF. But this is because SPEECH2TEXT takes a different spectrogram as input, one that is suboptimal for predicting neural responses to audio (S2 Fig, S5 Table). Similarly, untrained networks that take the waveform (rather than a spectrogram) as input will generally provide worse models of neural activity than the (optimal) STRF, because they are mostly linear in the waveform across layers. This can be seen for WAV2LETTER and WAV2VEC2 in our study (Fig 3) and for WAV2VEC2 in (e.g.) the recent study using ANNs as encoding models for the electrocorticogram [14].

Results in the literature that seem to contradict this ordering of encoding models are sometimes misleading. For example, a recent fMRI study in humans listening to speech [9] found that untrained WAV2VEC2 (which takes only a waveform as input) outpredicts the mel spectrogram (figure S3, *op. cit.*). But that spectrogram does not logarithmically scale the amplitudes,

which greatly reduces its predictive power (see again S2 Fig). More subtly, "untrained" models that are constructed by permuting the weights of trained models, rather than re-initializing them, can behave quite differently. Trained weights will typically be large, pushing outputs into saturating regions of some of the nonlinearities, losing information. This can degrade an ANN's performance as an encoding model well below the level of the STRF, even though the ANN takes the spectrogram as input [12]. For example, in S7 Fig, we re-run our analyses on the COCHRESNET50 from Tuckute and colleagues, using both re-initialization and permutation to generate untrained networks. For monkey vocalizations, in particular, the permuted network produces much worse predictions than the re-initialized network.

## The effect of the choice of ANN on encoding performance

Perhaps surprisingly, we did not observe systematic differences in model performance based on architectural choices. The best overall encoding model, for responses to both English speech and monkey vocalizations, was the first recurrent layer of DEEPSPEECH2. But the first self-attentional (transformer) layers of the WHISPER networks provide only slightly worse predictive power, particularly for speech. And the purely convolutional WAV2LETTER nearly matches DEEPSPEECH2's performance on monkey vocalizations. This last fact likewise shows that the performance of trained networks, in contrast to untrained networks, is not determined by the choice of input (waveform vs. spectrogram). Nor does network size seem to matter much, since the two WHISPER models are nearly identical as encoding models.

There is, however, some reason to believe that using more variable training data improves subsequent encoding performance. The two WHISPER networks and DEEPSPEECH2 make consistently strong predictions across stimulus types, and these are the networks trained on large, heterogeneous data sets. Indirect evidence is also provided by the fact that pre-training WAV2VEC2 on AudioSet [37] (a diverse set of sounds from internet videos) rather than LibriSpeech, and skipping fine-tuning altogether, boosts the encoding performance of deeper layers (S4 Fig). On the other hand, this does not improve the *peak* encoding performance of WAV2VEC2. We consider this to be an important avenue for future research.

## Speech as a stimulus and speech-trained ANNs as models

We chose speech-recognition for the pretext task because (1) we have found English sentences to elicit robust responses from neurons in squirrel-monkey core and belt; (2) monkey vocal calls contain somewhat similar spectrotemporal features; and (3) labeled training sets for English speech are several orders of magnitude larger than anything available for monkey calls or environmental sounds. Still, those more ecologically relevant sounds do lack the complex phonemic structure of English; and, conversely, contain sounds not found in English. From this perspective, it is not surprising that the *deepest* layers of the ANNs do not provide as good features for explaining the activity of neurons in the squirrel monkey. Furthermore, although we found nearly the same number of core multi-units tuned or well-tuned to speech as to monkey vocalizations, this ratio is much lower in non-primary areas (S3 Table). This is consistent with the observation that non-primary neurons are more specialized and sparsely firing. Similarly, the explanatory power of our ANNs is on average worse for neurons in non-primary rather than in primary areas. This suggests that we are still missing important features for explaining higher-order auditory cortex.

One way to address these issues would be to train ANNs *unsupervised* on monkey vocalizations or environmental sounds. However, comparing the results of such an experiment with the results of this study is not straightforward, because it is harder to evaluate what "good" performance is on the unsupervised task. For example, if such a network outperforms

the networks evaluated in this study, is it the result of better task performance or of a better task? We have therefore not taken this approach in the present study, although we consider it the clear follow up to this investigation.

## Correspondence between the hierarchies of ANNs and auditory cortex

We found a soft correspondence between (on the one hand) the hierarchy of the ANNs and (on the other) the distinction between primary and non-primary auditory cortex (Fig 5). A similar, although perhaps slightly less subtle distinction was found by Li and colleagues [14] in ECoG data—in this case between the homologous regions in humans, Heschl's gyrus and superior temporal gyrus, respectively (figure 2a, *op. cit.*). In fMRI studies [9,12], the correspondence is more conspicuous still. Although all of these studies are in humans, rather than squirrel monkeys, the emergence of more robust hierarchical distinctions at coarser spatial and temporal granularities suggests that single neurons are simply more diverse in their functional role. Averaging over time and space may mask this diversity.

## ANNs can predict the number of spikes in small bins

What the present study reveals, that cannot be observed from the electrocorticogram or fMRI, is that the representational correspondence with artificial neural networks emerges at the level of the single neuron, or at least the multi-unit (typically 1–5 neurons), not merely at the population level: The number of action potentials occurring in 20-ms intervals is well explained by features learned by the ANNs.

So ANNs can explain spiking activity in auditory cortex at time scales as fine as the networks' Nyquist limits. But can they explain neural activity at time scales finer still? After all, neurons in the primate auditory cortex are known to encode information about stimuli at almost millisecond precision [38].

Here we are limited by our networks and, presumably, the pretext task. The Nyquist limits of the ANNs are set by architectural choices (e.g., stride lengths of convolutions), which could in theory be changed. However, these choices were made to achieve optimal performance on the pretext task; in particular, by the intermediate layers, sampling rates decline to about 50 Hz, which is on the order of the phoneme-production rate (10–20 Hz). Thus, complex features that require multiple layers of nonlinearities but are at finer time scales will not be predictable by such models, and the models probably cannot be changed to predict them without sacrificing performance on the pretext task.

This is another reason to explore non-speech ANNs. An interesting alternative, however, would be use *sped-up* speech as stimuli. This would retain the same enormous, labeled datasets for training; increase the *optimal* sampling rates for the ANNs; and (arguably) decrease only minimally the relevance of the stimulus to the monkey.

## Biological implausibility of networks components

Transformer attention, bidirectional recurrence, and standard convolutions are all non-causal. In theory, all of these could be remedied (with causal masking, the use of unidirectional RNNs only, and causal convolutions, respectively), although these would have adverse affects on performance on the pretext task. We have made no attempt to address these here and consider this important future work. Similarly, none of the ANNs allow information to flow from deeper to earlier layers during processing, whereas primate cortex is known to have dense feedback connections from higher to lower areas of the processing hierarchy. A new class of

ANNs known as "predictive coding networks" has attempted to capture this aspect of neural computation [39]; they would make potentially very interesting alternatives to standard ANNs as encoding models for the cortex.

## Supporting Information

**S1 Text** Noise correction of correlations
(PDF)

**S2 Text** Pretrained networks
(PDF)

**S1 Table.** ANN training details.
(PDF)

**S2 Table.** Recording probes used in experiments.
(PDF)

**S1 Fig.** Noise correction and unit selection. A: Procedure for computing the true and null distributions of trial-to-trial correlations. Both distributions are computed using a set of stimuli with multiple presentations. For each stimulus in the set, a distinct pair of trials is randomly selected (STEP 1) and the responses concatenated in random order to form two response sequences, $U$ and $V$ (STEP 2). For computing the null distribution only, $V$ is circularly shifted by half its length (STEP 3), which preserves the marginal response statistics for each channel but destroys any possible temporal correlations with $U$. Finally, the correlation coefficient between $U$ and $V$ is computed (STEP 4). This process is repeated 100,000 times to obtain distributions (true and null) of correlation coefficients. B: Number of multi-units with at least $\delta$ standard deviations between their true and null distributions, as a function of delta. Results are nearly identical if the inter-mean (blue) or inter-median (orange) distance is used. C: True (blue) and null (gray) distributions of correlations for three randomly selected channels whose means (indicated by dotted vertical lines) are separated by a gap of $\delta = 0.1$, 0.5 and 1.0 times the standard deviation of the null distribution. For all three multi-units, the true distribution is shifted significantly to the right of the null distribution under a Wilcoxon rank-sum test ($p<0.05$), despite overlapping very heavily (see esp. $\delta = 0.1$).
(PDF)

**S3 Table.** Number of tuned neurons. See Materials and methods for criteria for being tuned and well-tuned. Stimulus classes are TIMIT sentences ("speech") and monkey vocalizations ("mVox").
(PDF)

**S4 Table.** ANN layer properties. RF: receptive field; Ts: sampling period.
(PDF)

**S5 Table.** Comparison of different spectrogram computations.
(PDF)

**S2 Fig.** Comparison of STRF-neuron correlations for various choices of spectrogram computation. Each box depicts the distribution of correlations across multi-units for a particular choice of spectrogram. The median, middle quartiles, and 5th/95th percentiles are indicated with a black line, shading, and whiskers (resp.). Results are shown separately for each stimulus type, speech and monkey-vocalization (vox).
(PDF)

**S6 Table.** ANN task performance. Comparison of pretrained models on speech recognition tasks, reported as word error rates (WER, %) across three test datasets. All test sets are distinct from the training data used for each model to ensure a fair comparison.
(PDF)

**S3 Fig.** Model-neuron correlation ratios. All subpanels show the correlations between model predictions and the multi-unit activity they are supposed to predict, each normalized by the corresponding STRF-neuron correlation. A: Model-neuron correlation ratios for speech (TIMIT) stimuli. Bar plot: median (across multi-units) correlation ratio for the STRF (gray, necessarily 1.0) and of the best layers of each of the neural networks, both trained (dark colors) and untrained (light colors). Line plots: the distributions of correlation ratios as a function of ANN layer for each of six trained networks (colored) and their untrained counterparts (gray). The median (solid line) and interquartile range (shaded region) are shown. The layer type is indicated along the top of each plot: convolutional (brown), self-attention (blue), and recurrent (light blue). B: The same as A but using monkey vocalizations for stimuli.
(PDF)

**S4 Fig.** Model-neuron correlations for two different trained versions of wav2vec2. All subpanels show correlations between model predictions and the multi-unit activity they are supposed to predict. A: Model-neuron correlations for speech (TIMIT) stimuli for wav2vec2, either trained as in the main text (yellow), or only pre-trained on AudioSet (red). B: The same as A but using monkey vocalizations for stimuli.
(PDF)

**S5 Fig.** Maximum (across layers) median ANN-neuron correlation vs. ANN word error rate on three ASR data sets. Each point corresponds to an ANN (color scheme as throughout). Circles: speech (TIMIT) stimuli; plus signs: monkey vocalizations.
(PDF)

**S6 Fig.** Distributions of most predictive layers (normalized) for primary (blue) and non-primary (orange) neurons. Histograms and corresponding kernel density estimates are shown as a function of network depth (from shallowest to deepest) for all neurons, separately for each ANN. A: TIMIT stimuli. B: Monkey-vocalization stimuli. For each pair of distributions, a Wilcoxon rank-sum test was used to determine if the non-primary neurons "prefer" deeper layers of the network (* for $p<0.05$, *
* for $p<0.01$, *
** for $p<0.001$).
(PDF)

**S7 Fig.** Model-neuron correlations for cochresnet50 with two different types of "untrained" model. All subpanels show correlations between model predictions and the multi-unit activity they are supposed to predict. A: Model-neuron correlations for speech (TIMIT) stimuli. The untrained networks (black line and shading) were created either by re-initializing the weights or by permuting the weights of the trained model. B: The same as A but using monkey vocalizations for stimuli.
(PDF)

**S8 Fig.** Model-neuron correlations using $\delta = 1.0$ as the inclusion criterion. All subpanels show correlations between model predictions and the multi-unit activity they are supposed to predict. A: Model-neuron correlations for speech (TIMIT) stimuli. Bar plot: median (across multi-units) correlation of the STRF (gray) and of the best layers of each of the neural networks, both trained (dark colors) and untrained (light colors). Line plots: the distributions of correlations as a function of ANN layer for each of six trained networks (colored) and their untrained counterparts (gray). The median (solid line) and interquartile range (shaded region) are shown. At starred layers, the trained ANN is significantly superior to its untrained counterpart (top row of stars) or a STRF (bottom row of stars; Wilcoxon signed-rank test with $p < 0.01$). To control the false discovery rate, we apply the Benjamini–Hochberg correction separately for each model. The layer type is indicated along the top of each plot: convolutional (brown), self-attention (blue), and recurrent (light blue). B: The same as A but using monkey vocalizations for stimuli.
(PDF)

**S9 Fig.** Distributions of model-neuron correlations as a function of maximum frequency of the model predictions using $\delta = 1.0$ as the inclusion criterion. Median (solid line) and interquartile range (shaded) are indicated. For each ANN, results are shown only for the layer that was most predictive at 50-ms bins. First, the layer's responses were low-pass filtered at the frequency indicated on the horizontal axis. Then a linear readout (TRF) was fit to predict spiking activity binned at 20 ms. The vertical axis shows the resulting distribution of correlations on the test set. The red star indicates the cut-off frequency yielding the largest median (across multi-units) correlation; black dots indicate frequencies yielding correlation distributions indistinguishable from that at the red star (Wilcoxon signed-rank test with $p < 0.01$).
(PDF)

**S10 Fig.** Distributions of most predictive layers (normalized) for primary (blue) and non-primary (orange) multi-unit activity using $\delta = 1.0$ as the inclusion criterion. Histograms and corresponding kernel density estimates are shown as a function of network depth (from shallowest to deepest), pooled across all neurons and all six ANNs. The distribution of preferred layers is significantly "deeper" for non-primary than primary neurons (Wilcoxon rank-sum test, $p < 0.001$).
(PDF)

## Acknowledgments

Some neural networks were trained on GPUs generously donated by the Nvidia Corporation.

## Author contributions

**Conceptualization:** Bilal Ahmed, Brian J. Malone, Joseph G. Makin.

**Data curation:** Joshua D. Downer, Brian J. Malone.

**Formal analysis:** Bilal Ahmed.

**Funding acquisition:** Joseph G. Makin.

**Investigation:** Joshua D. Downer, Brian J. Malone, Joseph G. Makin.

**Methodology:** Bilal Ahmed, Brian J. Malone, Joseph G. Makin.

**Project administration:** Joseph G. Makin.

**Resources:** Joseph G. Makin.

**Software:** Bilal Ahmed.

**Supervision:** Brian J. Malone, Joseph G. Makin.

**Validation:** Bilal Ahmed, Joseph G. Makin.

**Writing – original draft:** Bilal Ahmed, Joseph G. Makin.

**Writing – review & editing:** Bilal Ahmed, Joshua D. Downer, Brian J. Malone, Joseph G. Makin.

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
