## [Decision Letter · Decision Letter 0]

14 Apr 2025

PCOMPBIOL-D-25-00349

Deep neural networks explain spiking activity in auditory cortex

PLOS Computational Biology

Dear Dr. Makin,

Thank you for submitting your manuscript to PLOS Computational Biology. After careful consideration, we feel that it has merit but does not fully meet PLOS Computational Biology's publication criteria as it currently stands. Therefore, we invite you to submit a revised version of the manuscript that addresses the points raised during the review process.

Please submit your revised manuscript within 60 days Jun 14 2025 11:59PM. If you will need more time than this to complete your revisions, please reply to this message or contact the journal office at ploscompbiol@plos.org. Please include the following items when submitting your revised manuscript:

We look forward to receiving your revised manuscript.

Kind regards,

Jian Liu

Academic Editor

PLOS Computational Biology

Hugues Berry

Section Editor

PLOS Computational Biology

**Additional Editor Comments :**

The manuscript presents an interesting approach to modeling auditory cortex neuron responses using artificial neural networks and shows promising potential. However, several aspects require further clarification and development to strengthen the work. These include:

1) Validation of the results across different modeling settings to ensure robustness.

2) More detailed comparisons with related studies to better highlight the findings.

3) Clearer interpretation of the results, particularly their implications for understanding auditory cortex function.

4) Additional consideration of specific features from auditory cortex neurons that could refine the model's biological relevance.

Please also consider to revise the Data Availability Statement to make sure that all code is available via Github, and all data available via a public repository.

**Journal Requirements:**

3) We have noticed that you have uploaded Supporting Information files, but you have not included a complete list of legends. Please add a full list of legends for your Supporting Information files (Supplementary Tables) after the references list.

4) We notice that your supplementary figures are uploaded with the file type 'Figure'. Please note that they should be uploaded separately with the file type 'Supporting Information'. 

5) We notice that your supplementary Tables are included in the manuscript file. Please remove them and upload them with the file type 'Supporting Information'. Please ensure that each Supporting Information file has a legend listed in the manuscript after the references list.

Potential Copyright Issues:

i) Please confirm (a) that you are the photographer of 1, or (b) provide written permission from the photographer to publish the photo(s) under our CC BY 4.0 license.

7) When completing the data availability statement of the submission form, you indicated that you will make your data available on acceptance. We strongly recommend all authors decide on a data sharing plan before acceptance, as the process can be lengthy and hold up publication timelines. Please note that, though access restrictions are acceptable now, your entire data will need to be made freely accessible if your manuscript is accepted for publication. This policy applies to all data except where public deposition would breach compliance with the protocol approved by your research ethics board. If you are unable to adhere to our open data policy, please kindly revise your statement to explain your reasoning and we will seek the editor's input on an exemption. Please be assured that, once you have provided your new statement, the assessment of your exemption will not hold up the peer review process.

8) Please amend your detailed Financial Disclosure statement. This is published with the article. It must therefore be completed in full sentences and contain the exact wording you wish to be published.

3) If any authors received a salary from any of your funders, please state which authors and which funders.

9) Please ensure that the funders and grant numbers match between the Financial Disclosure field and the Funding Information tab in your submission form. Note that the funders must be provided in the same order in both places as well. Currently, the order of the funders is different in both places.

**Reviewers' comments:**

Reviewer's Responses to Questions

Reviewer #1: The authors present an interesting comparative study on different artificial neural network (ANN) architectures for speech recognition, and analyse similarity between ANN’s hidden unit activation and biological neurons in auditory cortex. This study follows from the successful works in visual cortex (by Yamins and Di Carlo 2014), yet applies this approach to auditory cortex. While there are interesting and novel results, I find positioning of the paper (with respect to related works) and interpretation of the results quite misleading, as I explain below point-by-point.

Line 10: “Heretofore, however, the correspondence has only been shown at gross scales, either temporal (∼1s) or spatial (1000s of neurons).” – There are no references provided. Which works do the authors refer to? Even the original [Yamins et al. 2014] study was performed on a fine time scale, 100ms windows taken 70ms after stimulus presentation, and at a single neuron resolution, which is already a counter-example to this statement.

Line 78 (and 523): ‘In particular, we asked how well sequences of spike counts in small (∼50-ms) bins can be (linearly) predicted by different layers of the ANNs.’ The 50ms bin is gigantic for electrophysiological studies. Most of the motor cortical studies in primates use 5ms or 20ms bins (see, e.g. datasets in a Neural Latents Benchmark); the works on decoding imaginary position from hippocampal activity also typically use 20ms (see works from Buzsaki lab). This is a typical behaviorally-relevant timescale, that is accessible with electrophysiological recordings. Even the [Yamins et al. 2014] paper uses 100ms window. Therefore, the claims that 50ms is a ‘small’ bin size must be corrected.

Line 97: ‘Electrodes consisted of either 16- or a 64-channel linear probes.’ – no type / geometry of the probe was provided. It would be useful to understand the limitation of the recording technique (and support claims like line 526 ‘single “unit” (typically 1–5 neurons)’)

Line 113: ‘Spikes were identified by threshold crossing’ – what was the threshold? What software was used?

Lines 120-160: The textual description of mathematical operations, instead of equations, lengthens and complicates the methods section. It is hard to decipher 1) what correlation was computed (correlation between what and what?); 2) how exactly was the model-neural correlation normalised;

On the content of these methods:

1. Why is sampling done ‘without replacement’ for estimating trial-to-trial correlation? Is it only suggesting that authors do not pick 2 identical samples for correlation? It seems likely that they still sample such pairs with replacement; otherwise, where would 100’000 assignments come from in line 130? If my understanding is correct, then it is “sampling non-identical pairs of responses with replacement”.

2. What are the ‘first’ and ‘second’ responses? Just 2 randomly drawn stimulus presentations?

3. Where does 1.6 seconds/sentence (line 127) come from? Were responses cut to the same length of 1.6s for this correlation analysis?

A figure/equation would be helpful to explain this procedure.

Line 158: ‘We estimated this correlation with the median of the trial-to-trial response’ – why the median? The correction was derived for the mean response.

Line 164: “generating (corrected) correlations much greater than 1.0. This makes all results noisier and harder to interpret.” – this alone is a weak justification for applying a stricter selection of units included in the analysis. Perhaps, if the mean correlation was used, the stability would have been better. Was mean > median trial-to-trial correlation for most of the neurons?

Line 228: How was resampling (upsampling) done for hidden layers?

Line 289: The correct term for thresholded activity on a recording channel is a MUA: multy-unit activity. Calling MUA a ‘neuron’ is misleading.

Line 330 (and also repeated in line 372): ‘Indeed, the most predictive ANN layers correlate with the median neuron at about 0.65 (noise-corrected). This is comparable to the best models of auditory neurons to date [21], which are complex neural networks fit directly to the neural data, whereas our models fit only the linear readout.’ – First, the results from [21] are not presented in this paper, are they? How can we compare [21] to the current paper if they are using a different noise correction (mean rather than median) and different datasets? To justify this claim, one would need to evaluate the model from ref. [21] on the same dataset and apply the same noise correction.

Figure 1: Impossible to understand what is on the axes, as not a single axis in the plots was labeled. Why are all scatter plots the same? Are these points taken from the data, or drawn randomly for illustration?

Figure 3: It would be useful to add linear decoding from all layers together. It could be that the recorded neural population spans multiple processing layers, so it would be useful to know how single layer prediction compares to all layers together. If all-layer prediction is not so far from the best-layer prediction, then it would also suggest that the layer-by-layer comparison approach, originally developed for mostly feed-forward visual cortex, can be applied to auditory cortex.

Reviewer #2: In the manuscript, the authors describe the application of deep neural networks to develop encoding models for neurons in the auditory cortex of squirrel monkeys. The findings indicate correlations between the model predictions and the activities of low-variability neurons, demonstrating that artificial neural networks (ANNs) can predict the fine temporal structure within spike-count sequences. Additionally, through the analysis of the most predictive layers for primary and non-primary neurons, they propose an anatomical hierarchy.

Below are my comments:

(1) Given the use of six presumably architecturally distinct deep neural networks for prediction, it remains unclear why these specific networks were chosen. A detailed comparison of the performance differences among these networks would provide valuable insight into the modeling outcomes.

(2) Beyond neuronal response, comparing additional features such as frequency tuning between neurons and models could offer further understanding.

(3) Concerning the hierarchy analysis, it's uncertain whether the hierarchical structure observed in ANN layers mirrors that of the auditory cortex. The current evidence does not strongly support this assertion.

(4) Clarification on which subregions of the auditory cortex were recorded from, and specifically which cortical layers, would be beneficial.

(5) More details on the training process of the ANNs, including hyperparameters used, would aid in replicating and understanding the study’s results.

(6) It is unclear if there is a clustering of neurons in Fig. 5. Additional explanation may help address this observation.

Reviewer #3: In the paper "Deep neural networks explain spiking activity in auditory cortex", the authors investigate the relationship between the activity of neurons in the auditory cortex of monkeys and the activity of artificial neural networks (ANNs) trained on auditory tasks, and untrained ANNs. Architectures of the ANNs include fully convolutional, recurrent, and Transformer-based networks, in the form of previously published architectures such as Wav2Letter, Speech2Text, Wav2Vec2, DeepSpeech2, and Whisper. The authors linearly predict spike counts of the auditory cortex neurons from the activity of ANN layers and find that auditory cortex spike counts are better predicted from trained ANN layers than from spectrotemporal receptive fields or untrained ANN layers. In particular, the authors highlight the unprecedented high temporal resolution for such an approach. The authors conclude that auditory cortex neurons are more tuned to intermediate features of deep neural networks than to the amplitude spectrum. The authors also find a weak correspondence between ANN hierarchies and areas of auditory cortex.

The paper is well-written and provides a clear and detailed description of the methods and results. The authors make a significant contribution to the field by demonstrating the potential of deep neural networks to explain spiking activity in the auditory cortex at fine time scales. The results provide new insights into the relationship between artificial and biological neural networks and highlight the potential of deep learning models to explain neural activity in the brain. That said, there are several major and minor concerns that should be addressed.

Major comments:

1. It seems the authors did not apply any multiple testing correction to the results. Given the large number of comparisons made in the study, especially in Fig. 3, it would be important to apply a correction such as controlling the false discovery rate to account for multiple comparisons. This would help to reduce the likelihood of false positives and provide a more accurate assessment of the statistical significance of the results.

2. In lines 133-143, the authors describe their approach to construct a null distribution of correlations. However, this distribution assumed Poisson firing and neglects any over- or under-dispersion. Did the authors check the Fano factors of the neurons? If the Fano factors are not close to 1, the null distribution may not be appropriate. It would be helpful to provide more details on whether it is robust to deviations from Poisson firing.

3. In line 231-232, the authors explain that a 350 ms window of ANN activity was used to predict the neural responses. However, this window is much longer than the typical response times of auditory cortex neurons. It would be interesting to see how the results change with shorter windows, closer to the response times of the neurons.

4. In lines 484-491, the authors suggest that the untrained networks are essentially linear transformations of the input. This is an interesting idea that is presented as speculation. However, the idea could be easily tested by comparing to random projections with the same dimensionality. It would be interesting to see if the performance is similar to the untrained networks.

Minor comments:

1. Fig. 3: The color code is not immediately clear. A legend would be helpful. Please also label the x-axes. Labeling each row might also help to make the figure easier to interpret.

2. Line 382: Fig. 3B -> Fig. 3A

3. Line 428: the the -> the

4. S2-S7 Tables: rho columns missing.

**Have the authors made all data and (if applicable) computational code underlying the findings in their manuscript fully available?**

Reviewer #1: **No: **Authors state that 'All code will be made available via Github, and all data available via a public

repository, upon acceptance of the manuscript.'

Reviewer #2: Yes

Reviewer #3: **No: **The authors promise to make these available upon acceptance.

PLOS authors have the option to publish the peer review history of their article (what does this mean?). If published, this will include your full peer review and any attached files.

Reviewer #1: No

Reviewer #2: No

Reviewer #3: No

**Figure resubmission:**
---

## [Decision Letter · Decision Letter 1]

2 Jul 2025

PCOMPBIOL-D-25-00349R1

Deep neural networks explain spiking activity in auditory cortex

PLOS Computational Biology

Dear Dr. Makin,

Thank you for submitting your manuscript to PLOS Computational Biology. After careful consideration, we feel that it has merit but does not fully meet PLOS Computational Biology's publication criteria as it currently stands. Therefore, we invite you to submit a revised version of the manuscript that addresses the points raised during the review process.

Please submit your revised manuscript within 30 days Sep 01 2025 11:59PM. If you will need more time than this to complete your revisions, please reply to this message or contact the journal office at ploscompbiol@plos.org. Please include the following items when submitting your revised manuscript:

We look forward to receiving your revised manuscript.

Kind regards,

Jian Liu

Academic Editor

PLOS Computational Biology

Hugues Berry

Section Editor

PLOS Computational Biology

**Additional Editor Comments:**

The revision has been significantly improved. However, a few minor issues still need to be addressed. See the details below.

**Journal Requirements:**

**Reviewers' comments:**

Reviewer's Responses to Questions

**Comments to the Authors:**

Reviewer #1: Authors have significantly improved the clarity of the paper in this revision, mainly by adding excellent schematics and improving figures. The newly added section about Noise corrections, however, has some contradictions and requires a revision:

1. Line 633: Model-neural correlations -- this is unusual terminology, which needs to be explained. It does make sense after reading the whole new section, but not at the beginning.

2. Line 636: 'Intuitively, trial-to-trial correlations involve two noisy variables, whereas model-neuron correlations involve only one (the model is deterministic); and so the latter is just the square root of the former.)' -- what does this sentence mean? Correlations always involve 2 noisy variables.

3. Line 639: 'The result is well known but for completeness we prove it here precisely' -- are there any references?

4. Line 642: '(see figure)' -- Figure number and caption missing.

5. If the model is deterministic (see p. 2 above), then sigma_u = sigma_x and all derivations make no sense, since correlations would be precisely 1.

Reviewer #2: This revised version demonstrates marked improvement and addresses the concerns I previously raised. Consequently, I have no further comments.

Reviewer #3: The authors provided a comprehensive and detailed revision that addresses all of my concerns. They now use the Benjamini-Hochberg procedure to control the false discovery rate, making the statistical significance of their results more rigorous. They also replaced their Poisson null distribution with a much more appropriate one that retains the marginal statistics of the neuronal responses. Furthermore, they looked into the effect of the window length for predicting neuronal responses, providing an additional figure to support their claims. Finally, they clarified their statements on linear transformations, again using an additional figure to confirm their claim. All of my comments have been addressed.

**Have the authors made all data and (if applicable) computational code underlying the findings in their manuscript fully available?**

Reviewer #1: **No: **Author state that All code will be made available via Github, and all data available via a public

repository, upon acceptance of the manuscript.

Reviewer #2: Yes

Reviewer #3: **No: **They promised to do so on acceptance of their manuscript.

PLOS authors have the option to publish the peer review history of their article (what does this mean?). If published, this will include your full peer review and any attached files.

Reviewer #1: No

Reviewer #2: No

Reviewer #3: **Yes: **Arno Onken

**Figure resubmission:**
---

## [Decision Letter · Decision Letter 2]

14 Jul 2025

Dear Prof. Makin,

We are pleased to inform you that your manuscript 'Deep neural networks explain spiking activity in auditory cortex' has been provisionally accepted for publication in PLOS Computational Biology.

Best regards,

Jian Liu

Academic Editor

PLOS Computational Biology

Hugues Berry

Section Editor

PLOS Computational Biology

The requested clarification has been addressed, and the manuscript is now suitable for publication.

Reviewer's Responses to Questions

**Comments to the Authors:**

Reviewer #1: The authors have addressed reviewers concerns.

The confusion came from the sentence: 'model-neuron correlations involve only one (the model is deterministic)', while now it is clear that authors in fact also assume observational noise, which is not part of their model and which introduces variability.

**Have the authors made all data and (if applicable) computational code underlying the findings in their manuscript fully available?**

Reviewer #1: **No: **see authors statement

PLOS authors have the option to publish the peer review history of their article (what does this mean?). If published, this will include your full peer review and any attached files.

Reviewer #1: No

---

## [Editor Report · Acceptance letter]

PCOMPBIOL-D-25-00349R2

Deep neural networks explain spiking activity in auditory cortex

Dear Dr Makin,

I am pleased to inform you that your manuscript has been formally accepted for publication in PLOS Computational Biology. Your manuscript is now with our production department and you will be notified of the publication date in due course.

With kind regards,

Anita Estes
